# Cerebral Small Vessel Diseases and Outcomes for Acute Ischemic Stroke Patients after Endovascular Therapy

**DOI:** 10.3390/jcm11236883

**Published:** 2022-11-22

**Authors:** Yixin Zhao, Yuye Ning, Lei Lei, Huijie Yuan, Hui Liu, Guogang Luo, Meng Wei, Yongxin Li

**Affiliations:** 1Stroke Centre, Department of Neurology, The First Affiliated Hospital of the Xi’an Jiaotong University, No. 277 Yanta West Road, Xi’an 710061, China; 2Department of Medical Imaging, The First Affiliated Hospital of the Xi’an Jiaotong University, Xi’an 710061, China; 3Biobank, The First Affiliated Hospital of the Xi’an Jiaotong University, No. 277 Yanta West Road, Xi’an 710061, China; 4Department of Cardiovascular Surgery, The First Affiliated Hospital of the Xi’an Jiaotong University, 277 Yanta West Road, Xi’an 710061, China

**Keywords:** cerebral small vessel diseases, acute ischemic stroke, endovascular therapy, outcome, magnetic resonance imaging

## Abstract

The correlation between cerebral small vessel disease (CSVD) and the outcomes of acute ischemic stroke (AIS) patients after endovascular therapy (EVT) remains elusive. We aimed to investigate the effect of combined white matter hyperintensities (WMH) and enlarged perivascular spaces (EPVS) as detected in magnetic resonance imaging (MRI) at baseline on clinical outcomes in patients with AIS who underwent EVT. AIS patients that experienced EVT were retrospectively analyzed in this single-center study. Using MRIs taken prior to EVT, we rated WMH and EPVS as the burden of CSVD and dichotomized the population into two groups: absent-to-moderate and severe. Neurological outcome was assessed at day 90 with a modified Rankin Scale (mRS). Symptomatic intracerebral hemorrhage (sICH), early neurological deterioration (END), malignant cerebral edema (MCE), and hospital death were secondary outcomes. Of the 100 patients (64.0% male; mean age 63.71 ± 11.79 years), periventricular WMHs (28%), deep WMHs (41%), EPVS in basal ganglia (53%), and EPVS in centrum semiovale (73%) were observed. In addition, 69% had an absent-to-moderate total CSVD burden and 31.0% had a severe burden. The severe CSVD was not substantially linked to either the primary or secondary outcomes. Patients with AIS who underwent EVT had an elevated risk (OR: 7.89, 95% CI: 1.0, 62.53) of END if they also had EPVS. When considering WMH and EPVS together as a CSVD burden, there seemed to be no correlation between severe CSVD burden and sICH, END, or MCE following EVT for AIS patients. Further studies are warranted to clarify the relationship between CSVD burden and the occurrence, progression, and prognosis of AIS.

## 1. Introduction

Stroke is the most common serious manifestation of cerebrovascular disease and the third leading cause of death in China [1]. Acute ischemic stroke (AIS) due to large vessel occlusion (LVO) increases mortality by approximately 4.5 times [2]. Multiple randomized trials have established endovascular therapy (EVT) as the standard of care for appropriate LVO-AIS patients, such as MR CLEAN [3], REVASCAT [4], SWIFT PRIME [5], DAWN [6], DEFUSE 3 [7], etc. However, there are still complications related to EVT, including symptomatic intracranial hemorrhage (sICH), early neurological deterioration (END), malignant cerebral edema (MCE), and death in the hospital [8], posing an immediate threat to life or leading to more time spent in an intensive care unit, higher medical costs, and slower rehabilitation. In order to improve the poor prognoses caused by MT-related complications in LVO-AIS patients, the baseline clinical and imaging factors or clinical outcome-related factors of LVO-AIS patients undergoing MT need to be evaluated.

Cerebral small vessel disease (CSVD) is a group of pathological processes caused by cerebral small vessels, reflexing the brain aging and causing various lesions, such as white matter hyperintensities (WMH), enlarged perivascular spaces (EPVS), lacunes, and cerebral microbleeds (CMB) [9]. Overall CSVD burden scores have been proposed to represent the influence of brain function [10]. Recent studies have shown that CSVD is more prevalent in AIS patients than it is in general people [11], and a severe CSVD burden is associated with an increased risk of sICH and poor outcome in AIS patients after intravenous thrombolysis (IVT) [12,13], which is considered a risk factor for AIS patients. However, the results of numerous studies concentrating on the relationship between CSVD and outcomes for AIS patients after EVT remain inconsistent and uncertain [14,15,16]. In view of these reports, the overall prognosis for LVO-AIS patients can be improved by further refining the eligibility criteria for EVT and identifying additional patients who can potentially benefit. Therefore, we assessed the association of combined WMH and EPVS between clinical outcomes in LVO-AIS patients after EVT.

## 2. Materials and Methods

### 2.1. Study Population

The study population comprises a total of 550 LVO-AIS patients treated with EVT from January 2019 to April 2022 at the Stroke Centre and Department of Neurology, the First Affiliated Hospital of Xi’an Jiaotong University. The preliminary pooled sample was retrospectively queried to identify adult patients meeting the following criteria: (1) age > 18; (2) LVO-AIS patients confirmed by CTA, MRA, or DSA and treated with EVT; (3) clinical data are available; and (4) baseline clinical MRI, routinely obtained before EVT, including axial T1- and T2-weighted, coronal fluid-attenuated inversion recovery (FLAIR), and diffusion-weighted. Patients were excluded for incomplete clinical data, inadequate quality MRI sequences, or having been discharged within 72 h. Clinical and radiological features were collected. A 90-day modified Rankin scale (mRS) score was assessed through in-person or telephone interviews. Finally, a total of 100 patients was retrospectively collected. The protocols involving human participants were reviewed and approved by the local ethics committee of the First Affiliated Hospital of Xi’an Jiaotong University (application number XJTU1AF2022LSK-282). The patients and participants provided written informed consent to participate in this study.

### 2.2. Clinical Variables

Baseline demographic information were retrieved as follows: age, sex, medical history, occlusion site, laboratory data, the National Institutes of Health Stroke Scale (NIHSS) score, and procedural endovascular variables. Successful recanalization after EVT was defined by a thrombolysis in cerebral infarction (TICI) score of 2b or 3 [17]. All patients were evaluated by an experienced neurologist in an acute setting.

### 2.3. Imaging Assessment

According to the standards for reporting vascular changes on neuroimaging (STRIVE) criteria [9], one seasoned neurologist and one radiologist evaluated the CSVD markers (WMH and EPVS) in MRIs without access to clinical information. EPVS were counted using a 5-level scale in the BG and CS reign: 0 = no EPVS, 1 = 1–10, 2 = 11–20, 3 = 21–40, and 4 = >40 [18]. We used only the highest values for both hemispheres. A score of ≥ 2 in the basal ganglia (BG) or centrum semiovale (CS) was regarded as severe EPVS. The sum of the BG and CS-EPVS scores was used to determine the total EPVS. We rated WMH using the Fazekas scale [19], and we the WMH instances divided into periventricular white matter hyperintensity (PWMH) and deep white matter hyperintensity (DWMH). We also retained the dichotomizations used in a previous similar study [20]. We considered the presence of either of the following as markers of severe CSVD: (1) (early) confluent DWMHs (Fazekas score of 2 or 3) or irregular PWMHs extending into the DWMHs (Fazekas score of 3); and (2) severe (>10) EPVSs in the BG [10,20,21]. We therefore built up an aggregate dichotomic score, splitting the population into two groups: (1) absent-to-moderate CSVD (Fazekas score of 1 or BG-EPVSs ≤ 10); (2) severe CSVD (Fazekas score of 2 or 3 or BG-EPVSs > 10).

### 2.4. Outcome Assessment

The mRS was assessed, blinded from the CSVD burdens, through in-person or telephone interviews with patients or their surrogates at 3 months after AIS. A poor outcome was defined as an mRS score of >2. Secondary outcomes included sICH, defined according to the European Cooperative Acute Stroke Study 2 (ECASS II) criteria as an NIHSS score 4 points higher than the value at baseline, with the lowest value in the first 7 days [22]; END (increase in NIHSS of ≥4 within 24 h) [23]; MCE (swelling causing midline shift) [24]; and death in the hospital. The researchers measuring these outcomes were blinded to the groups.

### 2.5. Statistical Analysis

Statistical analysis was performed with SPSS version 26.0 (Chicago, IL, USA). Data are expressed as means ± standard errors or medians (interquartile ranges) for continuous variables and as percentages for categorical variables. Between-group comparisons of means were performed using one-way analysis of variance. Non-normal data were evaluated by a nonparametric Kruskal–Wallis test. Between-group comparisons of categorical variables were evaluated by the χ^2^ or Fisher exact test. A Kaplan–Meier curve and binary logistic regression were used to investigate the independent risk factors of poor prognoses of AIS patients. For all statistical analysis, a *p*-value of < 0.05 was considered statistically significant.

## 3. Results

From January 2019 to April 2022, 550 patients were admitted to the emergency department of the stroke center, of which 25 patients chose conservative treatment, 145 patients only received intravenous thrombolysis (IVT), and 165 patients were assessed by CT. Therefore, 380 patients (mean age 65.27 ± 12.49 years; 64.5% male) experienced EVT, of which 118 were excluded because of unavailability, incomplete sequence, or poor quality of MRI. The flowchart of patient selection is depicted in Figure 1. Finally, a total of 100 patients met the inclusion criteria (mean age: 63.71 ± 11.79 years, 64% male).

Compared with those (*n* = 380) excluded from current analyses, the baseline characteristics of the patients were similar between the included and excluded patients (Table 1). There were no differences between the included and excluded patients in terms of clinical severity or medical history.

### 3.1. Imaging Markers and CSVD Burden

At baseline, 24% of the patients received IVT, all patients underwent EVT, 34% had internal carotid artery (ICA) occlusion, 52% had middle cerebral artery (MCA) occlusion, and 14% had basilar artery (BA) or posterior cerebral artery (PCA) occlusion. Among the 100 patients with LVO-AIS that experienced EVT, 11 patients were lost to follow-up and 22 patients died, accounting for 22.0% (22/100), and there were 41 patients with poor functional outcomes (mRS of > 2 at 90 days), accounting for 41.0% of the patients with LVO-AIS that experienced EVT after follow-up. The baseline characteristics of the study population and its stratification per functional outcome are detailed in Table 2.

There was almost perfect interrater reproducibility for the presence of individual CSVD imaging markers. PWMHs, DWMHs, EPVS-BG, and EPVS-CS were found in 28 (28%), 41 (41%), 53 (53%), and 73 (73%) patients, respectively. Overall, 69 (69%) and 31 (31%) patients had absent-to-moderate and severe CSVD burden, respectively, while the patients with severe CSVD burden were older (67.16 ± 10.7 versus 62.16 ± 12.0, *p* = 0.049) and more likely to have hypertension and a higher systolic blood pressure (SBP). There were no significant differences in the gender, vascular medical history, and outcomes between the absent-to-moderate CSVD and severe CSVD patients, and 41% had the primary outcome of mRS > 2 at 90 days, 33% had sICH, 18% had END, 14% underwent MCE, and 4% died during admission.

### 3.2. Association between CSVD Burden and Outcome

The distribution of mRS scores by CSVD burden is shown in Figure 2. A poor outcome (mRS score of 3–6) was achieved by 24 patients (40.9%) with absent-to-moderate CSVD and 17 patients (56.7%) with severe CSVD.

Early neurological recovery was observed in 59 of 100 patients (59.0%) with available data, 24 patients (24.0%) with absent-to-moderate CSVD, and 17 patients (17.0%) with severe CSVD. Table 3 shows the outcomes of patients with absent-to-moderate CSVD or severe CSVD, and 41% of the patients had a primary outcome of poor mRS, 33% had sICH, 18% had END, 14% underwent MCE, and 4% died during admission. Univariable analysis demonstrated that there was no statistically significant interaction between the CSVD burden and functional outcome (OR: 1.907, CI: 0.784, 4.642). A severe CSVD burden was not significantly associated with primary and secondary outcomes in univariate or multivariate regression models.

Considering the whole population, we then evaluated the association between individual CSVD imaging markers, including WMH and EPVS, and outcomes. In univariate regression analyses, the patients with EPVS-CS were significantly associated with an increased risk of END (OR: 7.89, 95% CI: 1.0, 62.53). However, PWMH, DWMH, or EPVS-BG were not associated with 90-day mRS, END, sICH, or MCE. All variables with *p*  <  0.1 in the univariate analysis were entered into a multivariate logistic regression with a backward procedure. In the multivariable analysis, after adjusting for covariates, the poor functional outcomes (mRS > 2) were associated with DWMHs (OR: 3.624, CI: 1.12, 11.69), and END continued to be more likely in patients with EPVS-CS (adjusted OR: 14.78, 95% CI: 1.36, 160.64). In addition, patients with PWMH, DWMH, or EPVS-BG were also associated with END (Table 4).

## 4. Discussion

The impact of CSVD burden in the setting of MT is still poorly documented and the results of available studies have often been controversial [15,25,26]. In this retrospective investigation, the CSVD burden, including WMHS and EPVS, determined by MRI in LVO-AIS patients prior to EVT was carefully evaluated. We discovered that EPVS-CS was substantially related to an elevated incidence of END in LVO-AIS patients following EVT, regardless of age, gender, or vascular risk factors. In contrast, no link was seen between severe CSVD burden and a bad outcome. Our results suggest that EPVS-CS may be a clinically significant measure of END risk in LVO-AIS patients with EVT. It may be important to stratify ICH patients who are more prone to poor clinical outcomes according to EPVS-CS burden.

Patients with LVO-AIS had a higher prevalence of CSVD, presumably because age is the primary risk factor for developing CSVD. Thus, CSVD may be more common when LVO-AIS is present as a concurrent illness, potentially leading to worse functional results [27]. Previous research on the association between EPVS and post-EVT outcomes was lacking. A meta-analysis found, however, that significant WMHs were linked to a poor functional result 90 days following EVT in LVO-AIS [28]. A recent study indicated a significant association between WMHs and poor functional outcomes after EVT [29]. Evidence suggests that EPVS is more common in stroke patients and associated with poor outcomes in AIS patients [30]. Growing evidence indicates that EPVS is related to deep intracerebral hemorrhage (ICH) [31,32,33]. The numerous CSVD imaging markers listed above reflect several facets of CSVD, each with a distinct topology and likely distinct pathogenetic pathways. They frequently coexist, and an overall image of these indicators may indicate the overall condition of the distal small artery/arteriole bed; hence, the overall CSVD burden scores are reported in earlier studies [10]. However, in this study, HWMHs were associated with an increased risk of END after being adjusted by other CSVD markers, but not with other outcomes. It is possible that newer generations of EVT devices with a higher recanalization rate are associated with a lower risk of poor outcomes [17,34]. Furthermore, our findings need to be interpreted with caution, given the limitations of this study. Most notably, the prevalence of sICH is very high (33%) compared to previous studies, such as the 4.4% (28/634) prevalence in HERMES, which was likely due to the differences in inclusion criteria. We included patients with longer onset-to-treatment times (14.48 ± 21.53 h vs. 195.5 (142–260) minutes for 632 patients in HERMES) [17].

We found that EPVS-CS was significantly associated with an increased risk of END in LVO-AIS patients after EVT. This discovery strengthens the case for the relationship between EPVS and cerebrovascular disease; however, it is still unknown if EPVS changes during the development of END. The PVS is a space established around small blood vessels in the brain that facilitates the removal of waste, the interchange of cerebrospinal fluid, and the regulation of both brain homeostasis and the immune system [35,36]. With advancing age and the development of disease, the PVS can enlarge to the point where it is apparent on an MRI scan as a fluid-containing round or a linear delineated structure, particularly in the white matter of the basal ganglia BG and the CS [37]. EPVS might reflect the extravasation of fluid across damaged small vessel walls, possibly compounded by the recruitment of inflammatory cells to the PVS, where they might promote further loss of blood–brain barrier (BBB) integrity and impair perivascular fluid transport [38], which may increase the risk of worsening functional outcomes [20]. In the current study, we found a relationship between EPVS-CS and an increased risk of END. In addition to impaired BBB, as mentioned above, which could aggravate cerebral edema early after AIS, long-term hypoperfusion of the microcirculation in severe CSVD could also accelerate infract expansion, both of which could lead to deterioration or fluctuations [39]. However, pretreatment collateral status is another important factor governing post-EVT functional outcome, which relies on numerous factors involving hemodynamic, metabolic, and neural mechanisms, but not microcirculation damage alone [25]. A previous study showed that CSVD is associated with impaired collateral recruitment [40]. Thus, EPVS may be associated with endothelial dysfunction and disruption of the blood–brain barrier, representing a different kind of vascular damage and having a different pathogenesis, which indicates a more severe marker of CSVD.

Overall, this study suggests that patients with a severe CSVD burden, who constitute a considerable proportion of AIS-LVO patients, would benefit from EVT, although neurologic deficits can fluctuate in the hyperacute stage. Therefore, severe CSVD burden should not be used as an exclusion index to dominate the clinical decision for EVT, which will enable more stroke patients to benefit from EVT. In the future, large studies are needed to verify our results.

We acknowledge the following limitations: First, we did not adjust our analyses for other MRI markers of CSVD, including cerebral microbleeds, lacunes, or brain atrophy. Second, the sample size was relatively small, but the trial would have sufficient power to detect a clinically significant difference in the primary outcome between those with absent-to-moderate and severe overall CSVD burden. Furthermore, an MRI exam at baseline was an inclusion criterion in this study, which is not a routine imaging workup for patients receiving EVT treatment. Patients with more severe stroke or worse general conditions were more likely to be unable to receive brain MRIs in the hyperacute stage. Yet, this was inevitable as an MRI was needed for the assessment of the CSVD imaging markers, particularly for EPVS that cannot be assessed with other imaging modalities. We compared the baseline characteristics of patients included in and excluded from the current analyses and found no differences in the clinical severity or medical history between the included and excluded patients. Last, but not least, the generalizability of the current findings needs further validation.

## 5. Conclusions

A significant majority of LVO-AIS patients exhibited severe CSVD load prior to EVT. Individual CSVDs are associated with an elevated risk of END, but a severe CSVD burden, as measured by an MRI, was not substantially associated with sICH, END, or MCE in LVO-AIS patients receiving EVT. Consequently, in our study, when considering WMH and EPVS together as radiological expressions of CSVD burden, it should not be an exclusion indicator when establishing patient eligibility for EVT. Further studies are warranted to clarify the relationship between CSVD burden and the occurrence, progression, and prognosis of AIS.

## Figures and Tables

**Figure 1 jcm-11-06883-f001:**
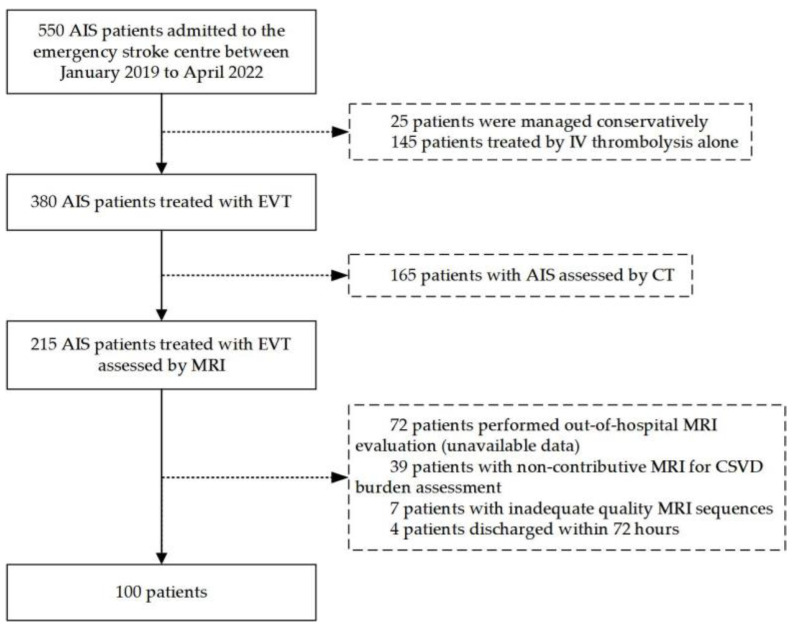
Flow chart of the study. AIS: acute ischemic stroke; EVT: endovascular therapy; IV: intravenous; CSVD: cerebral small vessel disease.

**Figure 2 jcm-11-06883-f002:**
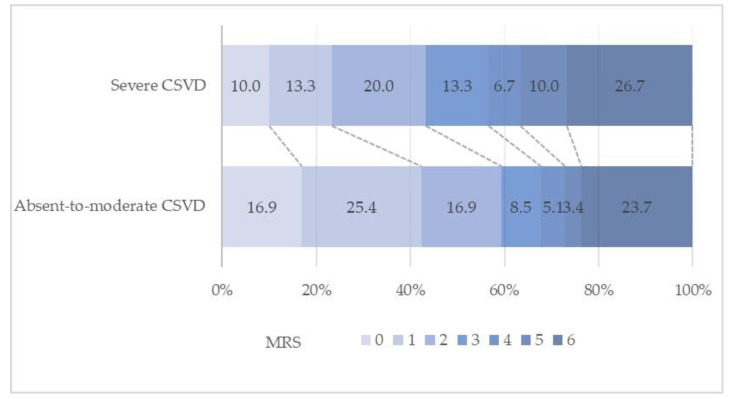
Distribution of 90 d modified Rankin scale (mRS) scores according to CSVD burden.

**Table 1 jcm-11-06883-t001:** Baseline characteristics of the included versus excluded patients.

	Total (*n* = 380)	Included (*n* = 100)	Excluded (*n* = 280)	*p*-Value
Age, years	65.27 ± 12.49	12.71 ± 0.76	11.79 ± 1.18	0.146
Gender, male, *n* (%)	245 (64.5)	181 (64.6)	64 (64.0)	0.904
Medical history, *n* (%)				
Hypertension	238 (62.6)	62 (62.0)	176 (62.9)	0.904
Diabetes mellitus	97 (25.5)	31 (31.0)	66 (23.6)	0.145
Coronary heart disease	75 (19.7)	22 (22.0)	53 (18.9)	0.559
Dyslipidemia	46 (12.1)	13 (13.0)	33 (11.8)	0.724
Atrial fibrillation	115 (30.3)	25 (25.0)	90 (32.1)	0.206
General status				
BMI, kg/m^2^	27.96 ± 34.36	34.56 ± 60.76	25.54 ± 15.79	0.234
Admission SBP, mmHg	142.64 ± 24.36	144.71 ± 23.09	141.92 ± 24.81	0.378
Admission DBP, mmHg	82.03 ± 15.17	83.19 ± 13.92	81.63 ± 15.60	0.429
IVT, *n* (%)	82 (21.6)	24 (24.0)	58 (20.7)	0.483
Admission NIHSS	11.43 ± 6.970	11.79 ± 6.147	11.29 ± 7.277	0.543
Onset-to-treatment, h	16.10 ± 24.02	14.48 ± 21.53	16.74 ± 24.95	0.427
Duration of treatment, min	125.89 ± 84.18	136.84 ± 125.53	121.56 ± 60.30	0.127
Laboratory examination				
WBC, cells/mL	10.02 ± 3.70	10.24 ± 3.68	9.94 ± 3.72	0.510
Hemoglobin, g/dL	107.85 ± 117.20	105.76 ± 116.32	108.58 ± 117.75	0.840
TG, mmol/L	1.16 ± 0.71	1.13 ± 0.61	1.17 ± 0.75	0.688
TC, mmol/L	3.87 ± 0.99	3.99 ± 0.90	3.82 ± 1.01	0.160
LDL, mmol/L	2.22 ± 0.80	2.28 ± 0.78	2.20 ± 0.81	0.425
HDL, mmol/L	1.01 ± 0.25	1.04 ± 0.28	1.00 ± 0.24	0.192
D-Dimer, mg/L	2.48 ± 4.64	2.02 ± 2.65	2.64 ± 5.15	0.301
Hcy, μmol/L	23.06 ± 15.91	24.05 ± 18.12	22.71 ± 15.08	0.503
Outcome, *n* (%)				
sICH	107 (28.2)	33 (33.0)	74 (26.4)	0.130

Values are means (SD) or medians (IQR) or numbers (%); Student’s *t*-tests were used for the comparison of continuous variables and χ^2^ tests were used for categorical variables. Abbreviations: BMI, body mass index; IVT, intravenous thrombolysis; WBC, white blood cells; TG, triglyceride; TC, total cholesterol; LDL, low-density lipoprotein; HDL, high-density lipoprotein cholesterol; Hcy, homocysteine; sICH, symptomatic intracranial hemorrhage.

**Table 2 jcm-11-06883-t002:** Baseline characteristics of the patients according to the overall CSVD burden.

	Absent-to-Moderate CSVD (*n* = 69)	Severe CSVD (*n* = 31)	*p*-Value
Age, years	62.16 ± 12.0	67.16 ± 10.7	0.049
Gender, male, *n* (%)	45 (65.2)	19 (61.3)	0.703
Medical history, *n* (%)			
Hypertension	37 (53.6)	25 (80.6)	0.010
Diabetes mellitus	23 (33.3)	8 (25.8)	0.452
Coronary heart disease	12 (17.4)	10 (32.3)	0.097
Dyslipidemia	8 (11.6)	5 (16.1)	0.533
Atrial fibrillation	16 (23.2)	9 (29.0)	0.533
General status			
BMI, kg/m^2^	24.94 ± 5.05	22.59 ± 3.51	0.147
Admission SBP, mmHg	142.23 ± 21.48	153.6 ± 21.7	0.033
Admission NIHSS	11.41 ± 6.13	12.65 ± 6.2	0.354
Laboratory examination			
WBC, cells/mL	9.37 ± 4.42	10.42 ± 3.95	0.275
Hemoglobin, g/dL	106.37 ± 117.0	129.27 ± 130.17	0.390
TG, mmol/L	1.12 ± 0.34	1.13 ± 0.74	0.926
TC, mmol/L	3.52 ± 1.02	3.67 ± 0.66	0.454
LDL, mmol/L	2.26 ± 0.83	2.21 ± 0.68	0.837
HDL, mmol/L	0.98 ± 0.27	1.14 ± 0.34	0.033
D-Dimer, mg/L	3.42 ± 10.13	2.13 ± 2.69	0.511
Hcy, μmol/L	21.22 ± 14.27	31.13 ± 23.49	0.016
Occlusion site			
ICA	25 (36.2)	9 (29.0)	0.482
MCA	36 (52.2)	16 (51.6)	0.959
PCA/BA	8 (11.6)	6 (19.4)	0.301
Treatment details and process times			
Onset-to-treatment, h	11.41 ± 14.30	15.12 ± 13.87	0.229
Duration of treatment, min	142.80 ± 114.19	123.13 ± 64.96	0.477
IVT, *n* (%)	17 (24.6)	7 (22.6)	0.824
Thrombectomy	56 (81.2)	22 (73.3)	0.381
Stent implantation	30 (43.5)	13 (43.3)	0.989
Balloon dilatation	27 (39.1)	14 (46.7)	0.484
contrast, ml	215.8 ± 101.12	243.0 ± 103.8	0.225
TICI 2b/3, *n* (%)	54(78.3)	24 (77.4)	0.925
Outcome, *n* (%)			
sICH	23 (33.3)	10 (32.3)	0.916
END	11 (15.9)	7 (22.6)	0.424
MCE	8 (11.6)	6 (20.0)	0.270
In-hospital death	5 (5.8)	0	0.178
mRS > 2	24 (40.7)	17 (56.7)	0.153

CSVD, cerebral small vessel diseases; BMI, body mass index; NIHSS, the National Institutes of Health Stroke Scale; WBC, white blood cells; TG, triglyceride; TC, total cholesterol; LDL, low-density lipoprotein; HDL, high-density lipoprotein cholesterol; Hcy, homocysteine; ICA, internal carotid artery; MCA, middle cerebral artery; BA, basilar artery; PCA, posterior cerebral artery; IVT, intravenous thrombolysis; TICI, thrombolysis in cerebral infarction; sICH, symptomatic intracranial hemorrhage; END, early neurological deterioration; MCE, malignant cerebral edema; mRS, the modified Rankin scale.

**Table 3 jcm-11-06883-t003:** Univariate and multivariate logistic regression for the outcomes by severe CSVD.

	Univariate Analysis OR (95%CI)	Multivariate Analysis		
		Model 1 OR (95%CI)	Model 2 OR (95%CI)	Model 3 OR (95%CI)
mRS > 2	1.907 (0.784, 4.642)	2.453 (0.869, 6.711)	2.656 (0.824, 8.557)	2.636 (0.780, 8.905)
sICH	0.952 (0.386, 2.352)	0.958 (0.365, 2.519)	1.036 (0.328, 3.265)	0.997 (0.313, 3.180)
END	1.538 (0.533, 4.440))	1.618 (0.520, 5.035)	1.911 (0.504, 7.242)	1.773 (0.460, 6.839)
MCE	1.906 (0.598, 6.075)	2.604 (0.713, 9.517)	2.036 (0.476, 8.716)	1.808 (0.408, 8.001)

CSVD, cerebral small vessel diseases; mRS, the modified Rankin scale; sICH, symptomatic intracranial hemorrhage; END, early neurological deterioration; MCE, malignant cerebral edema; OR, odd ratio. Model 1 was adjusted for demographics and vascular risk factors, including age, sex, history of hypertension. Model 2 was adjusted for variables in model 1, plus HDL and Hcy. Model 3 was adjusted for variables in models 1 and 2 and TICI scores of 2b/3.

**Table 4 jcm-11-06883-t004:** Associations between individual CVSD imaging markers and outcomes by logistic regression.

	mRS > 2		sICH		END		MCE	
	OR (95% CI)	Adjusted *	OR (95% CI)	Adjusted *	OR (95% CI)	Adjusted *	OR (95% CI)	Adjusted *
PWMHs	1.250 (0.50, 3.12)	2.981 (0.79, 11.32)	0.947 (0.37, 2.41)	0.853 (0.19, 3.98)	0.625 (0.08, 4.96)	0.402 (0.03, 5.30)	1.078 (0.31, 3.78)	1.2 (0.23, 6.23)
DWMHs	2.316 (0.98, 5.46)	**5.524 (1.49, 20.52)**	0.905 (0.39, 2.12)	0.468 (0.10, 2.25)	2.06 (0.73, 5.77)	2.273 (0.65, 7.93)	1.13 (0.36, 3.53)	1.11 (0.22, 5.49)
EPVS-BG	0.827 (0.36, 1.92)	1.073 (0.34, 3.36)	1.097 (0.48, 2.53)	2.61 (0.59, 11.52)	1.13 (0.41, 3.16)	0.473 (0.11, 2.0)	0.64 (0.2, 1.99)	0.52 (0.12, 2.28)
EPVS-CS	2.063 (0.78, 5.49)	1.024 (0.31, 3.43)	2.054 (0.734, 5.72)	3.893 (0.73, 20.73)	**7.89 (1.0, 62.53)**	**9.391 (1.13, 78.24)**	2.5 (0.52, 11.99)	3.71 (0.68, 20.22)

CSVD, cerebral small vessel diseases; PWMH, periventricular white matter hyperintensity; DWMH, deep white matter hyperintensity; BG, basal ganglia; CS, centrum semiovale; EPVS, enlarged perivascular spaces; mRS, the modified Rankin scale; sICH, symptomatic intracranial hemorrhage; END, early neurological deterioration; MCE, malignant cerebral edema; OR, odd ratio. Bold type indicates *p* < 0.05. Adjusted *: adjusted for demographics and vascular risk factors, including sex, history of hypertension and diabetes mellitus, NIHSS, onset-to-treatment time, and TICI scores of 2b/3.

## Data Availability

The data generated and/or analyzed during the current study are available from the corresponding author on reasonable request.

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
