# Peer review of "Cerebral Small Vessel Diseases and Outcomes for Acute Ischemic Stroke Patients after Endovascular Therapy"

_jcm, 2022, doi:10.3390/jcm11236883_

Round 1
Reviewer 1 Report
This a retrospective analysis of a group of patients, trying to understand the impact of CSVD markers on functional outcome after EVT.
It is relevant and adressess a hot topic on patient selection for EVT.
The introduction is short but sharp. I just have a brief comment on this part. The articles cited at line 43 (introduction) are not the most relevant; these would be perhaps the initial 5 big trials (REVASCAT; SWIFT PRIME; ESCAPE; MR CLEAN; EXTEND IA) and also the ones that have demostrated the role of MT after 6h (DAWN and DEFUSE 3)
The methods are well explained. There are, however, a few issues. First of all, there is no long term outcome (for example, modified Rankin Score at 3 months). Also, regarding sample: among 550 potential patients, only 100 were elegible for the study, which raises questions regarding representability
As for the results, they are clear. I just have one concern regarding the rate of symptomatic intracranial hemorrhage, which is quite high (for example, as compared to the rate reported ah the HERMES collaboration).
The discussion is comprehensive and supported by the results. I just have a brief comment on a specific sentence.
Line 199: it is said that "EPVS might develop in the presence of END." Well, that is not the case. EPVS might be associated with END but they don't develop after END, they were presente beferehand
T
Reviewer 2 Report
Dear Editor,
thank you for the opportunity to review this interesting manuscript.
Zhao and Coauthors aimed to investigate the correlation between cerebral small vessel disease and outcome in acute ischemic stroke patients treated with endovascular therapy.
The aim is clear and clinically meaningful as imaging factors related to outcome in AIS patients undergoing MT need further investigation, and, according to literature, CSVD is a plausible risk factor that can be associated with a worse outcome.
They retrospectively analyzed a cohort of consecutive patients with AIS who were treated with endovascular therapy. All patients included in the analysis had a baseline brain MRI.
Their main finding is that while the presence of severe CSVD is not associated with the outcomes of interest, single CSVD markers (PWMHs, DWMHs and EPVS-CS) were associated with early neurological deterioration.
The main strengths are the MRI based evaluation of CSVD in acute stroke patients, that make this paper novel and interesting, and the clinical relevance of the topic.
However, I have several concerns regarding this study:
- They screened 551 patients and included only 100 of them, about 18%. This is a significant limit that make the study cohort poorly representative. Why were all these patients excluded? The authors wrote “we excluded incomplete clinical data (n=443)”. However, considering the possible important effect of these exclusions on the results I think that should be stated in the paper how many patients were excluded for lacking MRI, that I think is the main reason.
Moreover, included and excluded patients should be compared regarding baseline characteristics and outcomes.If absence of baseline MRI is the leading cause of exclusion, a worrisome selection bias can be present. The choice of MRI over TC in a subgroup of patients is probably motivated by different clinical and radiological characteristics that can also affect outcome.
They partially reported this limit in the discussion, but, in my opinion, it can have a profound impact on the results and should be better dealt with.
- The prevalence of sICH is very high (33%) compared to previous literature. Again, this is due to a selection bias? sICH prevalence is the same in excluded patients? The neurologic severity is not reported. Moreover, time from onset to recanalization and ASPECT score are not reported; these two parameters could help to better depict the study population and maybe explain the high rate of sICH. If the Authors have other hypothesis, they should discuss this point.
- Covariates used in the second model (Table 3) are not specified. As 2/3 of the variables that are significantly associated with the outcome in the adjusted model are not in the univariate logistic regression, knowing the other explanatory variables included is mandatory to allow the reader to interpreter the provided results.
Overall, the statistical analysis performed, particularly how they built the regression models (particularly table 3) and how they chose the covariates should be better described.
- In the discussion the Authors stated: “This discovery strengthens the case for the relationship between EPVS and cerebrovascular disease, however it is still unknown why EPVS might develop in the presence of END”. From their study, they cannot say that EPVS develop in the presence of END. First, they were already present before END (as scored on the baseline MRI). Second, not having a short-term follow-up MRI, they don’t know if EPVS changes during the development of END. I’m also doubtful that EPVS can develop acutely.
If the authors have also early MRI follow-up data this point can be discussed, otherwise, I think that this sentence should be made clearer, because maybe I did not understand it as they intended.
- The Authors should discuss why, in their opinion, EPVS-CS are strongly associated with END, while EPVS-BG are not. They think that they represent different kind of vascular damage and have different pathogenesis? EPVS-CS represent a more severe maker of CSVD? EPVS-CS could be in some why a cause of END?
The same should be done for PWMHs e DWMHs. The positive association between DWMHs and END can be understood. But how they comment on the significative negative association between PWMHs and END.
Without at least a plausible hypothesis that explain these results, considering the limits of the study design, the results themself loss a lot of significance.
- “…all patients underwent EVT, 38% had Internal carotid artery (ICA) occlusion, 52% had middle cerebral artery (MCA) occlusion.”. The remaining 10%? Were posterior circulation strokes included or ACA occlusions? Or maybe it is just a typing error…
Round 2
Reviewer 2 Report
The authors addressed all of my previous observations and the manuscript was revised accordingly. Some limitations remain, but are now clearly indicated in the manuscript. No further revisions are needed, in my opinion.